# Antifungal Effect of Chitosan/Nano-TiO_2_ Composite Coatings against *Colletotrichum gloeosporioides*, *Cladosporium oxysporum* and *Penicillium steckii*

**DOI:** 10.3390/molecules26154401

**Published:** 2021-07-21

**Authors:** Yage Xing, Rumeng Yi, Hua Yang, Qinglian Xu, Ruihan Huang, Jing Tang, Xuanlin Li, Xiaocui Liu, Lin Wu, Xingmei Liao, Xiufang Bi, Jinze Yu

**Affiliations:** 1Key Laboratory of Grain and Oil Processing and Food Safety of Sichuan Province, College of Food and Bioengineering, Xihua University, Chengdu 610039, China; a17380301029@163.com (R.Y.); yang1hua1@yeah.net (H.Y.); xuqinglian01@163.com (Q.X.); huangr_h@163.com (R.H.); TJ19980809@163.com (J.T.); lxl0519@126.com (X.L.); xiaocuiliu777@126.com (X.L.); wlin0702@163.com (L.W.); xingmeiliao123@163.com (X.L.); bxf1221@163.com (X.B.); 2Key Laboratory of Food Non Thermal Technology, Engineering Technology Research Center of Food Non Thermal, Yibin Xihua University Research Institute, Yibin 644004, China; 3Tianjin Key Laboratory of Postharvest Physiology and Storage of Agricultural Products, Tianjin 300384, China; 13032291162@126.com

**Keywords:** chitosan, nano-TiO_2_, mold

## Abstract

Postharvest pathogens such as *C. gloeosporioides* (MA), *C.*
*oxysporum* (ME) and *P. steckii* (MF) are the causal agents of disease in mangoes. This paper presents an in vitro investigation into the antifungal effect of a chitosan (CTS)/nano-titanium dioxide (TiO_2_) composite coating against MA, ME and MF. The results indicated that, the rates of MA, ME and MF mortality following the single chitosan treatment were 63.3%, 84.8% and 43.5%, respectively, while the rates of mycelial inhibition were 84.0%, 100% and 25.8%, respectively. However, following the addition of 0.5% nano-TiO_2_ into the CTS, both the mortality and mycelial inhibition rates for MA and ME reached 100%, and the mortality and mycelial inhibition rate for MF also increased significantly, reaching 75.4% and 57.3%, respectively. In the MA, the dry weight of mycelia after the CTS/0.5% nano-TiO_2_ treatment decreased by 36.3% in comparison with the untreated group, while the conductivity value was about 1.7 times that of the untreated group, and the protein dissolution rate and extravasation degree of nucleic acids also increased significantly. Thus, this research revealed the potential of CTS/nano-TiO_2_ composite coatings in the development of new antimicrobial materials.

## 1. Introduction

Fungal diseases caused by *Colletotrichum gloeosporioides* (MA), *Cladosporium oxysporum* (ME) and *Penicillium steckii* (MF) are a major cause of mangoes postharvest deterioration in mangoes, significantly affecting the quality and shortening the shelf-life of this popular fresh produce [1,2,3]. Mangoes grow in subtropical and tropical areas with high levels of humidity and high temperatures and are easily infected by the microorganisms that thrive equally well in such climates [4]. Mangoes are susceptible to rapid aging and postharvest rot brought about by such microorganisms, among which the anthracnose caused by MA is most serious. MA is known to cause obvious latent infection in mango tissues, including black or irregular spots on leaves; sunken black spots or necrosis lesions on petioles, stolons and fruits; and wilting of the whole plant due to anthracnose [5]. At present, chemical fungicides are the main method for the control of mango anthracnose, and carbendazim and other benzimidazole fungicides have been reported to have positive effects [6]. However, the long-term use and abuse of fungicides have increased pathogen resistance to fungicides and produced a serious worldwide health problem in farm animals and humans [7,8]. Therefore, a safer, more effective and long-lasting method of control is needed.

In recent years, edible coatings have been widely studied for their potential in the preservation of fruits and vegetables [9]. CTS, in particular, is well-known for its antibacterial, antioxidant and good food preservation properties. Used in the coating treatments of fruits and vegetables, it can form an acidic microenvironment around their surfaces, reducing the volatilization of moisture and restraining the respiratory rates of fresh produce [10]. Hamidreza showed that different kinds of CTS can have various bacteriostatic effects on bacteria and fungi. It was found that CTS could inhibit the growth of Gram-positive and Gram-negative microorganisms, most notably in that research inhibiting the growth of *Aspergillus flavus* and *Aspergillus niger* [11]. Huang’s in vitro studies showed that CTS significantly inhibited the growth of mycelia and spore germination in *Phytophthora*, reducing its resistance to various adverse conditions, and cooperated with pesticides, thereby potentially reducing the need for excessive input of chemical pesticides that are known to be toxic to human health [12]. However, the high viscosity of CTS solutions and its poor solubility in neutral conditions limit its practical application [13].

Nano-titanium dioxide (TiO_2_) is a highly active inorganic nano material, which exhibits qualities such as being non-toxic, antibacterial, promoting bacterial decomposition, anti-ultraviolet (UV) and super lipophilic [14,15,16]. It is, thus, widely used in the preparation of cosmetics [17], antibacterial fibers [18] and other fields. Under UV light, the electron holes in nano-TiO_2_ interact with water and oxygen to generate active oxygen species, especially hydroxyl radicals, thereby achieving antibacterial effects [19]. This sterilization method is simple, convenient and does not pollute the environment [20]. Moreover, nano-TiO_2_ has been particularly investigated due to its high tendency to form aggregates and, therefore, its low capacity to homogeneously disperse in organic media [9]. Researchers often use sodium dodecyl sulfate (SDS) [21], sodium laurate [22] and other surfactants to modify nano-TiO_2_ particles to improve their dispersion in the carriers [23]. Maneerat and Hayata [24] evaluated the antifungal effect of TiO_2_ in a coating film and found not only that the development of *Penicillium* rot in apples was significantly retarded by its photocatalytic reaction, but also that it decreased brown lesions and *Penicillium* rot infection in lemons. Chawengkijwanich and Hayata [25] demonstrated that nano-TiO_2_ coated film could reduce the risk of microbial growth on fresh-cut produce and, if incorporated into an edible film, could help to maintain their quality.

Chitosan/nano-TiO_2_ composite films are widely reported to have good antimicrobial activity [26] and superior properties that can be applied in aspects such as extending freshness, adsorption, degradation of organic pollutants, textile material modification, medical materials and other intelligent materials [27]. Soltaninejad presented a new, efficient photocatalyst (PVA/TiO_2_/CTS/Chl), in which the zones of inhibition for *Staphylococcus aureus* and *Escherichia coli* bacteria were around 2.08 and 1.98, respectively [28], and the investigation of Han [29] indicated that the killing rate of *E. coli* treated with CTS (TiO_2_ content 0.125%, CTS concentration 0.5%) was as high as 90% within 2 h. Furthermore, Arain [30] reported that the antibacterial effect of an AgCl-CTS-TiO_2_ composite was better than that of AgCl-TiO_2_ antibacterial material, and that when the concentrations of CTS and AgCl-TiO_2_ were 4 g/L and 10 g/L, respectively, antibacterial rates against *S. aureus* and *E. coli* reached 100%. However, little has been reported about the antifungal properties and mechanism of CTS/nano-TiO_2_ composite coatings.

Thus, the objective of this study was to explore the antifungal properties and mechanism of CTS/nano-TiO_2_ composite coatings against MA, ME and MF via in vitro testing. The mortality of mold, inhibition of mycelial growth, dry weight of mycelium, conductivity, protein dissolution rate and exosmosis of mold nucleic acids following different treatments in various media were determined.

## 2. Materials and Methods

### 2.1. Materials

MA, ME and MF were provided by the Laboratory of Food Biotechnology, School of Food and Bioengineering, Xihua University. CTS (deacetylated ≥95%) was purchased from Jinan Haidebei Marine Bioengineering Co. Ltd. (Jinan, China), and nano-TiO_2_ (30 nm) was purchased from Zhoushan Mingri Nanophase Material Technology Co., Ltd. (Zhejiang, China). PDA, PDB were purchased from Beijing Aobo Star Biotechnology Co., Ltd. (Beijing, China). Other chemicals and reagents were purchased locally and were of analytical grade. 

### 2.2. Coating Preparation

The CTS films were prepared according to the method described by Rhim with minor modifications [31], in which 1 g of CTS powder was dissolved in 100 mL of 1% (*v*/*v*) aqueous acetic acid solution with 1 g glycerin, then heated at 90 °C for about 20 min. The membrane solution was then filtered through eight layers of cotton cloth, and ultrasonicated for 30 min to complete preparation of the CTS solution (CTS). Thereafter, 0.1 g, 0.3 g and 0.5 g sodium laurate modified nano-TiO_2_ were dissolved as separate solutions in 1 g glycerin, followed by the sequential addition of 100 mL 1% aqueous acetic acid solution [32] and, finally, 1 g CTS powder. The remaining steps were the same as those employed during the preparation of the pure CTS film. In all, three treatments were investigated, namely CTS-0.1%, CTS-0.3% and CTS-0.5%. Lastly, 2 mL membrane films of different concentrations were mixed with 18 mL PDA medium, sterilized at 121 °C for 25 min and then cooled for later usage, to prepare media of different concentrations. PDA medium with an equal volume of sterile water was prepared for use as the control.

### 2.3. SEM Analysis

The surface morphology of the composite film was observed by SEM. The samples were dried and fixed on the stainless steel stage with conductive double-sided tape, the samples were observed under 10 kV accelerating voltage after sputtering and spraying gold.

### 2.4. Determination of Mold Mortality

Under aseptic conditions, 1 mL of the mold spore suspension (concentration approximately 10^6^ CFU/mL) was removed from the media containing different concentrations of composite coating, cultured at 28 °C for 72 h and then counted [33]. Each treatment was performed three times. Mortality rates were determined via the following formula [34]:

Mortality = ((number of mold colonies in control sample) − (number of viable bacteria in treated samples))/(number of mold colonies in control sample) × 100%

### 2.5. Inhibition Rate and Dry Weight of Mycelium

Under aseptic conditions, thriving mycelium with a diameter of 4 mm was placed in a conical flask containing potato dextrose broth (PDB) and subsequently cultured at 28 °C for 72 h. Each treatment was performed three times. The diameter of the colony was measured using the cross-sectional method, with the following formula [35]: 

Inhibition rate = (diameter of control colony − diameter of treated colony)/(diameter of contrast colony) × 100%

Diameter of colony = total diameter of colony − initial diameter of colony (4 mm)

To determine the dry weight of the mycelium, under aseptic conditions, a thriving mycelium colony with a diameter of 4 mm was inserted into a conical flask containing PDB. The cake was incubated on a shaking-table for 6 days (28 °C, 150 r/min), then centrifuged at low temperature (8000 r/min, 20 min), the supernatant was discarded, and the precipitates were dried to a constant weight. Each treatment was performed three times.

### 2.6. Determination of Conductivity, Protein Dissolution and Nucleic Acids Exosmosis

With reference to the method described by Zeng, a 10 mL sporozoa suspension (10^6^ CFU/mL) was placed in a conical flask, to which was added 2 mL different membrane solutions, and the mixture was cultured on a shaking table (28 °C, 150 r/min) [36]. After 10 h of sampling and centrifugation, the supernatant was removed to determine conductivity. Each treatment was performed three times.

As described by Xu, 10 mL sporozoa suspension (10^6^ CFU/mL) was placed in a conical flask, to which was added 2 mL different solutions, and then cultured in a shaker (28 °C, 150 r/min) [37]. After 10 h, sampling and centrifugation were conducted to determine the absorbance value of the supernatant at 280 nm for the protein dissolution rate and the absorbance value of the supernatant at 260 nm for the rate of nucleic acids exosmosis. Each treatment was performed three times.

### 2.7. Statistical Analysis

All statistical analyses were conducted using SPSS 20 and Origin 9.0. Data are presented as mean ± standard errors in the figures. All data obtained in this study were subjected to analysis of variance (ANOVA) followed by Duncan’s multiple range test to determine significant differences among the means at an α = 0.05 level.

## 3. Results

### 3.1. SEM Analysis

The morphology of CTS/nano-TiO_2_ nanocomposites observed by SEM was beneficial in evaluating the effects of the composite synthesis process. It can be seen from Figure 1a that the pure chitosan coating is a continuous, smooth and flat single homogeneous system. Figure 1b–d shows that the surface morphology of CTS/TiO_2_ composite film changes obviously with the addition of different concentrations of nano-TiO_2_, which is mainly due to the small bumps formed by nano-TiO_2_ particles embedded in chitosan, which makes the surface of the composite film uneven. When the TiO_2_ concentration consisted of 0.1% TiO_2_, some agglomerates were noticeable on the surface. With 0.3% TiO_2_, more agglomerates covered the surface of the film, and became coarse. However, when the concentration of titanium dioxide is 0.5%, the particles are evenly dispersed in the composite membrane system.

As shown in Figure 1, TiO_2_ particles were not uniformly distributed in the chitosan. Dong reported that when the content of nano-TiO_2_ is up to 30%, the distribution of nano particles in composite solution was uneven, which made the surface of the composite film rough after drying [38]. Furthermore, Hu [39] showed that some TiO_2_ particles were exposed on the fiber surface or gathered between the fibers with the increase of TiO_2_ content. The agglomeration may be due to the difficulty in dispersing the powdered nano-titanium dioxide in the viscosity solution [9]. In the preparation of the composite membrane, it is necessary to research the style of stirring and time in order to avoid the agglomeration of nanoparticles in the coating carrier.

### 3.2. Effect on the Mortality of Mold and Inhibition Rate of Mycelial Growth

The effects of different concentrations of the composite membrane solution investigated in this study on the mortality of mold are shown in Figure 2a. It is evident that, with increases in the nano-TiO_2_ content in the membrane solution, the mortality of mold after treatment also increased and tended to be stable. After being treated with only 1% CTS membrane solution, the mortality of MA was 63.3%; however, when the concentration of nano-TiO_2_ was 0.1% in the composite membrane solution, mold mortality increased significantly to 82.7% (*p* < 0.05), while with nano-TiO_2_ content of more than 0.3% the mold mortality stabilized at around 100%. Similarly, mold mortality in the ME was 84.8% after treatment with a single membrane solution of CTS, but increased significantly to 100% (*p* < 0.05) after treatment with the CTS/0.3% nano-TiO_2_ composite membrane solution. In the MF, mortality was 43.5% after treatment with the single CTS membrane solution, but increased significantly (*p* < 0.05) in line with the increase of nano-TiO_2_ in the composite membrane solution. When the content of nano-TiO_2_ in the composite membrane solution was up to 0.5%, the mortality reached 75.4%. Following treatment with single membrane solutions of CTS, the mold mortality rate, from highest in the ME and lowest in the MF. When the content of nano-TiO_2_ in the composite membrane solution was 0.3%, the mortality rates in the MA and ME were significantly higher than that in the MF. Furthermore, the effects of different concentrations of the CTS/nano-TiO_2_ composite membrane solution on the growth inhibition rate of the mold mycelia are shown in Figure 2b. As can be seen, with the increase in the nano-TiO_2_ content in the membrane solution, the inhibition rate of mycelial growth in the MA and MF after the membrane solution treatment also increased continuously. Following treatment with the CTS single membrane solution, the inhibition rates of mycelium growth in the MA and MF were 83.4% and 25.8%, respectively, with a significantly higher inhibition rate in the MA (*p* < 0.05). When the content of nano-TiO_2_ in the composite membrane liquid was 0.5%, mycelium growth inhibition rate was 100% in the MA, but just 57.3% in the MF. The inhibition rate of mycelium growth in the ME treated with single CTS membrane solution was as high as 100%, which was significantly higher than that of both the MA and MF treated with the same solution (*p* < 0.05).

Different fungi exhibit different sensitivities to CTS compound solutions, and the inhibition rates of CTS compound solutions with the same concentration are also different. In this study, the composite membrane solution was most effective in inhibiting mold on the ME. When the nano-TiO_2_ content was increased to 0.3%, the composite membrane solution could completely inhibit mycelial growth on the ME, followed by its rates of mycelial inhibition in the MA and then the MF. There have been some studies on the antibacterial effects of CTS and TiO_2_, they were shown in Table 1. Researchers have proposed some inferences on the mechanism of CTS in inhibiting fungi. CTS, for example, as a crab mixture, can integrate and remove the metal ions needed for microbial growth. Thus, it can inhibit the growth of the bacteria [40]. The growth rate of fungal seedling silk is affected by the intracellular calcium concentration [41]; however, as CTS can inhibit calcium and other nutrients, it, thus, limits the growth of fungi. Furthermore, it has been believed that CTS directly destroys the function of the cell membrane, thus exhibiting antibacterial properties. Since chitinase in pathogenic fungi can be overexpressed when the CTS concentration is high, this leads to the degradation of chitin, an important component of the fungal cell wall, thus destroying the cell wall of the pathogen and achieving the antifungal effect [42,43]. However, the antibacterial mechanism of nano-TiO_2_ nanoparticles is a consequence of the indirect reaction between the nano-TiO_2_ and cells. That is to say, the reaction of photogenerated electrons, photogenerated holes with water or dissolved oxygen in water forms reactive oxygen species (ROS), such as hydrogen oxygen free radicals and hydrogen peroxide free radicals, and produces chemically active hydroxyl groups and hydroxyl groups through redox reaction supercations, among others. The mechanism for bactericidal activity of TiO_2_ nanoparticles is shown in Figure 3. Active hydroxyl groups and supercations can react with biological macromolecules, such as lipids, proteins and enzymes, thereby directly damaging or destroying the structure of biological cells, such as bacteria, through a series of oxidation chain reactions, to achieve an antibacterial effect [44,45]. In addition, the mechanism involved in the antibacterial activity of a CTS/TiO_2_ composite may be related to the surface area of the catalyst. The CTS/TiO_2_ composite can adsorb well on the cell surface and penetrate the cell wall easily, thereby causing intracellular material leakage and impaired nuclear function, resulting in cell growth inhibition or death [46]. Similarly, the oxygen free radicals produced can attack the outer membrane, DNA, RNA and lipids, and oxidize or damage bacteria, leading to cell growth inhibition or death [47,48].

### 3.3. Effects of CTS/Nano-TiO_2_ Composite on Dry Weight of Mycelium

The effects of different concentrations of the CTS/nano-TiO_2_ composite membrane solutions on the dry weight of mold mycelia are shown in Figure 4. For MA, the dry weight of mycelium in the control group was 50.40 mg, which decreased by 16.3% to 42.20 mg after treatment with the single membrane solution of CTS and, with 0.3% nano-TiO_2_ in the composite membrane solution, decreased further to 33.50 mg, a reduction of 20.61% compared with that of the single CTS membrane treatment (*p* < 0.05). For ME, the dry weight of the mycelium in the control group was 38.30 mg, decreasing by 9.7% to 34.60 mg after treatment with a single membrane solution of CTS. With 0.3% nano-TiO_2_ in the composite membrane solution, the dry weight of the mycelium was 28.90 mg, a reduction of 16.5% compared with that of the single CTS membrane treatment (*p* < 0.05). For MF, the dry weight of the mycelium in the control group was 33.70 mg, decreasing by 7.4% to 31.20 mg after treatment with the single CTS membrane solution, and by a further 12.8% to 27.20 mg after treatment with 0.3% nano-TiO_2_ in the composite membrane solution (*p* < 0.05).

Thus, it is evident that with the increase of nano-TiO_2_ content in the composite membrane solutions, the dry weight of mold mycelium decreased continuously, stabilizing when the nano-TiO_2_ content exceeded 0.3%, which indicates that the inhibition effect of composite coatings on mycelium growth is more effective than that of a CTS coating alone. Cheng determined the effects of in vitro microwave and ultraviolet mutagenesis on the dry weight of mold mycelium and reported that the dry weight of treated *Trichoderma T-YS* was significantly higher than that of the control, indicating that such radiation treatments can have significant effects on the growth of *Trichoderma T-YS* strains, which is closely linked to the dry weight of their mycelia [54]. It may be that a CTS/nano-TiO_2_ composite membrane solution acts on the cell wall of mold to cause changes in its components, leading to the leakage of cell contents and, ultimately, cell ablation, in which its own enzymes catalyze the decomposition of chitin, and protein, nucleic acid, ammonia, free amino acids, organic phosphates and organic sulfur compounds are released, resulting in the decrease of mycelium dry weight [55].

### 3.4. Effect on the Conductivity of Mold

The effects of different concentrations of the CTS/nano-TiO_2_ composite membrane solution in this study on the electrical conductivity of mold are shown in Figure 5. The initial conductivity values of the MA, ME and MF were 236 uS/cm, 233 uS/cm and 230 uS/cm, respectively; however, after the molds had been treated with the CTS membrane solution, their conductivity values were found to have increased significantly to 370 uS/cm, 341 uS/cm and 354 uS/cm, respectively. Moreover, following the increase of nano-TiO_2_ content in the membrane liquid treatment, the conductivity of mold in the treatment group was significantly higher than that in the control group (*p* < 0.05), but stabilized in all three molds when the nano-TiO_2_ content reached 0.3%, with no further significant changes.

The plasma membrane is the permeable barrier of bacteria, playing an important role in regulating the concentrations of sodium, potassium and calcium plasma both inside and outside the cell, regulating the energy metabolism and material transportation of cells, and maintaining the stability of the intracellular environment [56]. The conductivities of MA, ME and MF treated with different concentrations of CTS or CTS/nano-TiO_2_ composite membrane solution were found to be significantly higher than those of the control group, increasing further still with the increase in the amount of nano-TiO_2_. Similarly, Li’s study showed that the relative conductivity of *E. coli* and *S. aureus* increased to 30% and 25%, respectively, after a 24 h treatment with CTS [57]. CTS can change the permeability of a mold cell membrane, resulting in extensive electrolyte leakage and, consequently, cell death [58,59]. While the antibacterial effect of nano-TiO_2_ is not specific, it is known to depend mainly on the strong oxidation of ROS produced by light excitation [60]. Nano-TiO_2_ requires only weak ultraviolet light (like sunlight on a cloudy day or fluorescent lamp light) to produce strong oxides such as hydroxyl radicals (•OH) and ROS, and •OH can destroy the unsaturated bonds in organic molecules, damage cell components, break the cell wall membrane, destroy the permeable barrier that helps to maintain cell activity, cause contents to overflow, generate H_2_O_2_ to enter the cell, and attack the internal functional components of the cell, ultimately resulting in cell death [41,61,62]. This further explains the increase in conductivity when nano-TiO_2_ content is increased in the composite treatment.

### 3.5. Effect on Protein Dissolution and Determination of Nucleic Acids Exosmosis 

The effects of the CTS/nano-TiO_2_ composite membrane solution at different concentrations on the rates of protein dissolution in the treated molds are presented in Figure 6a. In the MA, ME and MF the initial absorbance values were 0.282, 0.246 and 0.203, respectively, which increased 0.351, 0.273 and 0.329, respectively, after the single CTS membrane and composite membrane liquid treatments (*p* < 0.05). Nano-TiO_2_ is known to increase the rate of fungal protein dissolution, thereby changing membrane permeability. With the increase of nano-TiO_2_ content in the membrane solution, the light absorption value also increased continuously and significantly in all treated molds (*p* < 0.05). Because CTS and its composite membrane liquid can destroy the integrity of the cell membrane, thereby increasing its permeability, a large amount of protein is subsequently dissolved in the cell, and the light absorption value, measured at 280 nm, is also increased. 

Furthermore, the effects of different concentrations of the CTS/nano-TiO_2_ composite membrane solution on nucleic acid extravasation in the treated molds are shown in Figure 6b. While the initial absorbance values of the MA, ME and MF were 0.208, 0.225 and 0.233, respectively, after CTS treatment, these were found to have increased to 0.250, 0.247 and 0.251, respectively. Thus, compared with the control group, the absorbance values of the mold suspensions could be significantly increased by treatment with both the single and compound CTS membrane solutions (*p* < 0.05). With the increase of nano-TiO_2_ content in the membrane solution, the light absorption value also increased continuously and significantly (*p* < 0.05).

The leakage of cell contents may reflect serious and irreversible damage to a cell membrane. When this essential cell barrier is destroyed by bacteriostatic agents, the macromolecule material normally present in the cell is partially discharged to the extracellular solution, with this outflow including mainly intracellular protein and nucleic acid. The UV absorption peak of nucleic acid has a maximum absorbance value of OD_260_ nm, and the protein concentration is linearly related to the absorption value at OD_280_ nm. Therefore, by measuring the absorbance values of a mold’s solution at OD_260_ nm and OD_280_ nm after coating treatment, we can understand the release of macromolecular substances in mold [63]. In this study, the absorption value of the MA treated by the membrane solution with 0.3% nano-TiO_2_ content was significantly increased compared to that with a 0.1% nano-TiO_2_ content. In the MF, when the nano-TiO_2_ content was 0.5%, the light absorption value also underwent a significant change, indicating that the treatment of coating liquid affected the integrity of the cell membrane and increased the exudation of intracellular material. Zhang investigated the antibacterial activity of cinnamaldehyde against *S. aureus* and *E. coli* by observing the integrity and permeability of cell membrane, examining the changes in membrane potential and bacterial morphology via scanning electron microscopy (SEM). The experimental results showed that the shape of the two pathogenic bacteria were distorted, and that the integrity of the cell membrane was damaged, from which a large number of nucleic acids and proteins had leaked out, consequently hindering the normal growth of the bacterial cells [64]. Hence, it is speculated that the CTS/nano-TiO_2_ composite membrane solution destroyed the integrity of the pathogen cell membrane, thereby increasing its permeability, inducing leakage and the loss of intracellular substances, thus leading to cell death.

## 4. Conclusions

CTS was found to exert distinct and certain different antifungal effects on MA, ME and MF. The antifungal effects on MA and ME were stronger than those on the MF, suggesting the varying sensitivities of different fungi to antifungal agents. Compared with the CTS coating alone, the CTS/nano-TiO_2_ composite coating showed a better antifungal effect, restraining the growth of the mycelium, destroying cell membrane integrity, inducing the extravasation of intracellular protein and nucleic acid, increasing the conductivity value of the fungal suspensions, and killing the molds. These results, thus, demonstrate the potential of CTS/nano-TiO_2_ composite coatings in the development of novel and increasingly effective antimicrobial materials.

## Figures and Tables

**Figure 1 molecules-26-04401-f001:**
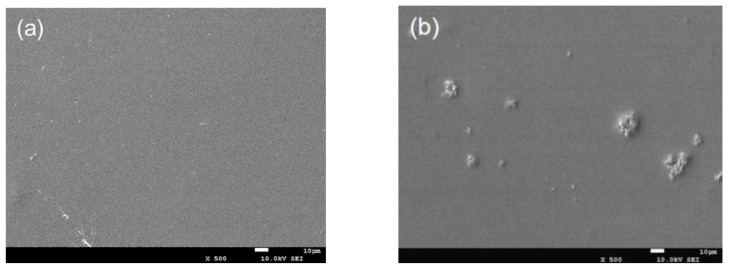
SEM image of pure chitosan film and different content of nano-TiO_2_ composite film (ck: pure chitosan coating film; (**a**–**d**): the chitosan composite coating film containing 0.1 g, 0.3 g and 0.5 g of modified nano-TiO_2_ respectively).

**Figure 2 molecules-26-04401-f002:**
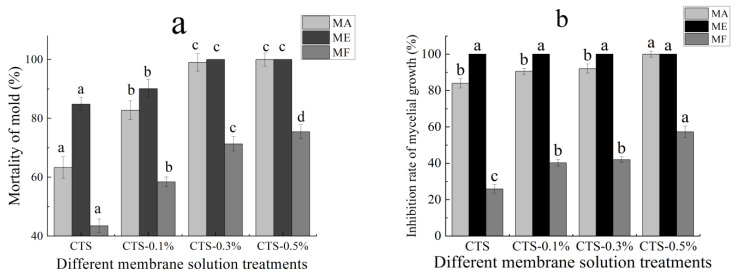
Effect of different concentrations of composite membrane solution on the mortality of mold (**a**) and inhibition rate of mycelial growth (**b**). The bars on the columns indicate standard deviation. Different letters indicate significant differences (*p* < 0.05) within an experiment (marked with letters).

**Figure 3 molecules-26-04401-f003:**
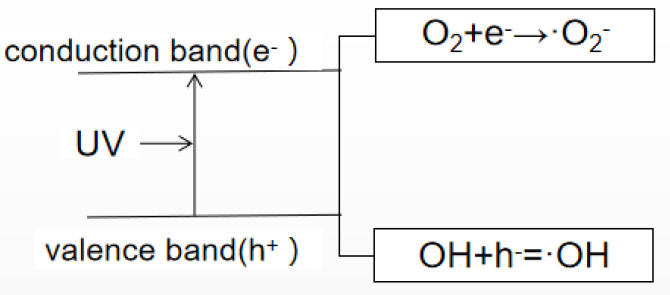
Photocatalytic principle of TiO_2_ nanoparticles.

**Figure 4 molecules-26-04401-f004:**
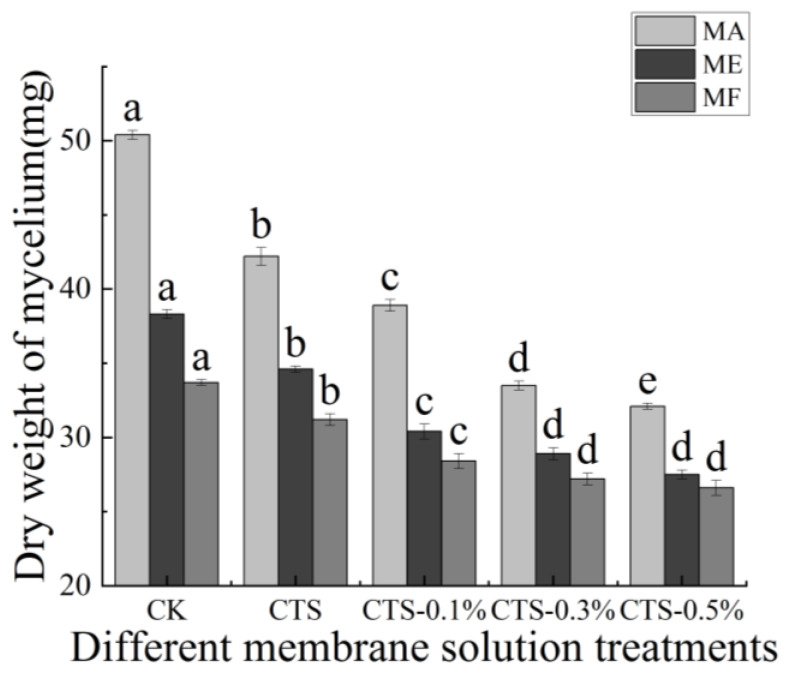
Effect of different concentrations of composite membrane solution on dry weight of mycelium. The bars on the columns indicate standard deviation. Different letters indicate significant differences (*p* < 0.05) within an experiment (marked with letters).

**Figure 5 molecules-26-04401-f005:**
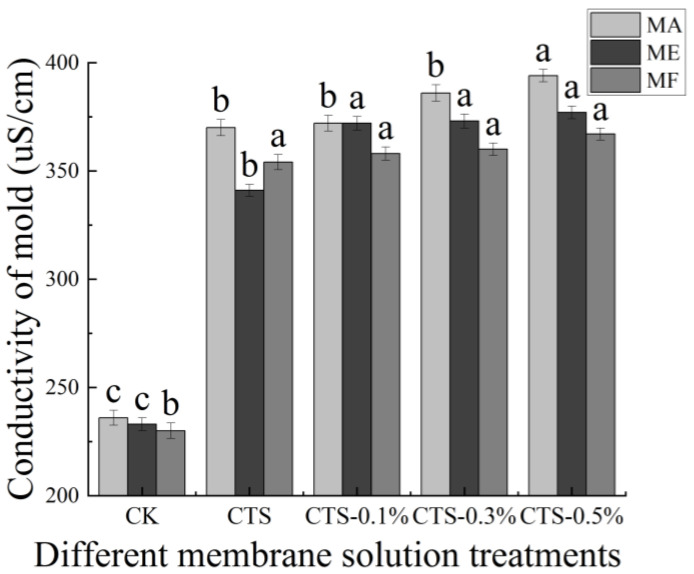
Effect of different concentrations of composite membrane solution on conductivity of mold. The bars on the columns indicate standard deviation. Different letters indicate significant differences (*p* < 0.05) within an experiment (marked with letters).

**Figure 6 molecules-26-04401-f006:**
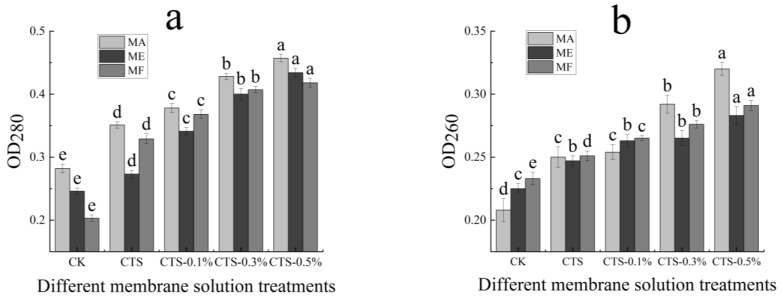
Effect of different concentrations of composite membrane solution on the protein dissolution rate (**a**) and determination of nucleic acids exosmosis (**b**) of mold. The bars on the columns indicate standard deviation. Different letters indicate significant differences (*p* < 0.05) within an experiment (marked with letters).

**Table 1 molecules-26-04401-t001:** Research on antimicrobial activity of chitosan or TiO_2_.

Fungi	Treatment	Antimicrobial Effect
*Fusarium graminearum*	100 μg/mL chitosan	Inhibition rate, 97.39% [49]
*P. Steckii*, *A. oryzae*	Red and blue light combined with CTS/TiO_2_ treatment for 120 h	Zone of inhibition, 4.7 mm, 54.8 mm [50]
*Phytophthora infestans*	0.05 g/L chitosan for 12 h	Spore germination rate, 1.29% [12]
*T. Viride*, *P. citrinum*	ZnTB and CuTB ((Cu, Zn)/TiO_2_ film-coated) for 14 and 28 days	Inhibition efficiency, 100% [51]
*C. albicans*	Fungi were treated with CTS/TiO_2_ at 37 °C for 24 h	Zone of inhibition, 14.66667 ± 0.5773 mm [52]
*B. maydis*	CTS/TiO_2_ treatment for 1 h	Inhibition rate, 100% [53]

## Data Availability

The data presented in this study are available on request from the corresponding author.

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
