# Peer review of "Antifungal Effect of Chitosan/Nano-TiO2 Composite Coatings against Colletotrichum gloeosporioides, Cladosporium oxysporum and Penicillium steckii"

_molecules, 2021, doi:10.3390/molecules26154401_

Round 1
Reviewer 1 Report
Main remarks
1. Lines 92 – 93
The authors wrote that the aim of the study “was to explore the antifungal properties and mechanism of CTS / nano-TiO2”. In my opinion, the authors do not study the mechanism of the CTS / nano-TiO2 impact on fungi, but rather use the mechanisms described in the literature to explain obtained results. There is also nothing in the Conclusions about the antifungal mechanism of CTS / nano-TiO2. Therefore, the aim of the work should be written a bit differently.
2 Lines 65-67
The authors discuss the interaction of UV radiation with TiO2 and the formation of active oxygen species or hydroxyl radicals. I think it would be good to write a reaction diagram between UV, TiO2 and O2.
In the Results chapter, the Authors use this reaction several times to discuss the results.
3. Lines 191 – 194
The obtained results were compared only with the results from one publication [37]. It is worth comparing the results of your research with the results of other authors, even if the experimental conditions were different. Such a comparison can be made, for example, in a table.
4. Lines 104 – 148. 4. Materials and Methods
It was not explained how many times each experiment was repeated. Since the standard deviation was given, I think several times, but how many? This information must be given.
5. Lines 138 – 148
It is written “With reference to the method described by Zeng,… [35],”
- Zeng, R. Study on antibacterial active ingredients, antibacterial mechanism and antiseptic effect on citrus. Nanchang University,2012.
and
“As described by Xu, 10 mL bacterial suspension (10-6 CFU/mL) was placed in a conical flask, to which was added 2 mL different solutions, and then cultured in a shaker (28°C, 150 r/min) [36].”
Do you study antibacterial or antifungal properties?
The methodology of experiment should be written as it was done.
6. Lines 19, 20, 160, 178
The authors gave their results with an accuracy of 0.01% (e.g. Abstract lines 19, 20). Were the measurements made with such precision? Maybe it is better to present the results with an accuracy of 0.1% as in line 194.
7.
The manuscript concerns the study of molds and antifungal effects but not bacteria and antibacterial effects. The authors use the terms bacteria, antibacterial, bacteriostatic, etc. This is an inappropriate. e.g. Conclusions lines 334-35. It is written “CTS was found to exert distinct and certain different antibacterial effects on MA, ME and MF. The antibacterial effects on MA and ME were stronger…”
In Abstract is written lines 17-18: “this paper presents an in vitro investigation into the antifungal effects and mechanism of a chitosan (CTS)/nano-titanium dioxide (TiO2) composite coating against MA, ME and MF.”
Please review the manuscript very carefully and correct any such inaccuracies.
8. Figures
The letters in the pictures are too small. I also propose to use other patterns in bars that are more different from each other.
There are letters a, b, c, d, e written above the standard deviation lines (in the top of bars). I have not found any information about what the letters mean!
Figure 1a. The scale on the 0Y axis should be split differently e.g. 60, 80, 100. Such change will make the results shown in Figure 1a easier to read.
9. Line 311 - instead of "bacteriostatic" it should be "Zeng [35] describes in manuscript antibacterial effect, while Xu [36] describes antifungal activity” - in the description of the methodology of the reviewed work there is an erroneous reference to the works of these authors, besides why describe the methodology for bacteria, since molds were the subject of the research - please correct the methodology in p. 2.5 and specify methodology that was used by the authors fungistatic"
Other comments
- In the title, I propose to give the full Latin names of the mold
- Lines 16, 56, 93 - in vitro - in italics
- Line 73 - Instead of "antibacterial" there should be “antifungal"
- Line 85 - E. coli - in italics
- Lines 120, 139, 144 Instead of 10-6, you would get 10-6
- Lines 123, 124 - Instead of “viable bacteria”, there should be “mold colonies”
- Line 138, 148, 286, 309 - instead of “nucleic exosmosis” should be “nucleic acids exosmosis”
- Line 172 - MF was entered twice - it should be "MA and ME were significantly higher than that in the MF"
- Line 204 - instead of "bactericidal effect [40,41]" it should be "antifungal" or "antimicrobial" - reference 40 relates to the antifungal activity of chitosan
- Line 316 - instead of bacteria "by measuring the absorbance values of a bacteria solution" should be "molds"
- Line 334, 335, 336, 337, 338 - instead of "antibacterial" it should be "antifungal"
- Line 339 - the word 'substances' was used unnecessarily
- Line 340 - instead of "bacterial suspensions" it should be "fungal s…"
- Lines 101, 110, 320 - there is TiO2 should be TiO2
- Line 115 It is "PDA" should be “potato dextrose agar (PDA)”
Reviewer 2 Report
Dear Editor,
The manuscript (molecules-1283390) reports about antifungal properties of a well-studied antimicrobial coating material composed of TiO2 incorporated Chitosan. In general, there are several published studies on the interaction of the same composition with bacteria and uncovering the antibacterial mechanism. This study seems to be very similar to those mentioned, with the difference in the microorganism. My major comments regarding this submission include:
1- Materials: there are a number of solvents and chemicals used in the process of synthesis whose producer's details and purity grade have not been included in "Materials".
2- Page 3, line 120, CFU value seems incorrect.
3- Page 3, line 123, is there any reference for the formula?
4- Figure 1, what a, b, and c labels on each column indicate? Not to mix up with the figures' labels use other indicators.
5- It is not clear how chitosan and TiO2 form a composite system, are the TiO2 nanoparticles coated with a chitosan layer or there is a chitosan film loaded with TiO2 nanoparticles? a Schematic could clarify the system configuration.
6- Considering the generation of reactive oxidative radicals by TiO2, I was wondering if such reactive agents do not harm (degrade) chitosan itself?
7-Page 6, Effects of CTS/nano-TiO2 composite on dry weight of mycelium; in general the results shown in graphs are exactly reported/repeated in such sections. What is the importance of stating the numbers/values represented by the graphs?
8- Page 7, Conductivity unit should be written with S capital.
9- There is no visual elements elaborating the synthesized system in terms of morphology, distribution of TiO2 nanoparticles, the graphs are unicolor and visually unattractive.
Author Response
Please see the attachment

This manuscript is a resubmission of an earlier submission. The following is a list of the peer review reports and author responses from that submission.